# Genetic Diversity, Analysis of Some Agro-Morphological and Quality Traits, and Utilization of Plant Resources of Coriander (*Coriandrum sativum*) Supported with Cluster and Multivariate Analyses

**DOI:** 10.3390/biology13110866

**Published:** 2024-10-24

**Authors:** Abdurrahman Basol, Gulsum Yaldiz, Mahmut Camlica

**Affiliations:** Department of Field Crops, Agriculture Faculty, Bolu Abant İzzet Baysal University, 14280 Bolu, Türkiye; abdurrahmanbasol@hotmail.com (A.B.); mcamlica25@gmail.com (M.C.)

**Keywords:** coriander, morphology, yield, fixed oil, essential

## Abstract

Genetic diversity of the different origin genotypes is one of the most important topics to evaluate the desired properties and select high-yield genotypes. Coriander (*Coriandrum sativum* L.) is an annual plant native to the Mediterranean region, Western Europe, and Asia that belongs to the Apiaceae (Umbelliferae) family. The fruits and essential oils of coriander are used for spice, folk remedies, perfumery, food, tobacco, soft and alcoholic beverages, and pharmaceutical industries in the different parts of the world. The yield and some quality characteristics of coriander genotypes of different origins should be investigated for the breeding program. In this study, both phenotypic, morphological, and yield values showed wide variations. Also, some analyses, such as cluster, heat map, and PCA analyses, revealed important results for the coriander genotypes.

## 1. Introduction

Coriander (*Coriandrum sativum* L.) is a medicinal and aromatic plant belonging to the Apiaceae family. It is cultivated in different parts of the world, and about 2.3 million tons of anise, badian, coriander, cumin, fennel, and juniper fruits were produced, along with 2.3 million ha in 2021 in the world, and 204 tons of coriander cultivation was produced in 2022 in 157.1 ha of land in Türkiye [1,2]. Between 2019 and 2021, India, Colombia, China, Saint Lucia, Mali, Russia, and Ghana exported the most coriander in the world, respectively. Coriander exports from Türkiye have been over USD 46.7 million in 2022 [2].

Coriander has a high commercial value in the global market, which is crucial for both the national economy and export opportunities. This requires the production of high-quality products that meet established standards. To grow such quality products, it is essential to select appropriate varieties and employ effective cultivation techniques tailored to the specific ecological conditions.

Dried fruits of coriander are used widely as a condiment, flavoring in sauces, liquor, cocoa, chocolate industries, and meat, bakery, and confectionery products. In addition, it is a mixed major ingredient with many curry powders as a standard matter. The aroma and flavor of coriander fruits are attributable to the essential oil present in oil glands in the mericarp. It contains up to 1%*v*/*w* essential oil content (EOC), and the main essential oil composition classified as monoterpenoid is linalool. Traditionally, coriander fruits are used to cure bed colds, spasmolytic, digestive, galactagogue, and stomach disorders and are also used as a drug for indigestion, against worms, rheumatism, and pain in the joints [3]. In addition to the medicinal uses, the essential oils of coriander are used in the flavoring of many food products and soap production. Moreover, this plant is superior to other oils of its class, an advantage due to being more stable and retaining its agreeable odor longer [4].

Climate change, a global concern, also affects coriander production due to irregular rainfall, increased water demand, and increased biotic and abiotic stresses. However, these changes increase CO_2_ concentration, thus increasing photosynthesis and accelerating the ripening process. To prepare for climate change, it is necessary the identification of heat, flooding, and drought-tolerant genotypes and the development of nutrient-efficient cultivars. Therefore, it will be essential to develop coriander genotypes that are resistant to stresses such as drought, moisture stress, salinity, and high temperatures. This requires high-priority research to address the impact of climate change. Based on all these explanations, our strategy in this study is to conduct adaptation studies with coriander genotypes from different countries and compare them with existing local genotypes/cultivars.

Fruit yield of plants is affected by some genetic factors connected to the environment. Furthermore, the coriander’s fruit composition was different under different ecological conditions and applications. The essential oil content and its compositions changed in different maturation periods, different parts of the coriander, different climatic conditions, geographical regions, agronomical applications, and genetic (genotype) factors [5,6,7,8]. In addition, fixed oil acids showed variability during fruit maturation periods, maturity stages, and geographical areas [9,10]. Thus, breeding studies on plant production are necessary to construct on the less affected by the environment for suitable selection criteria [11].

In recent years, there has been a clear upward trend in studies on coriander. There are studies on coriander around the world, but research on the agricultural and quality characteristics of coriander populations from different origins is not yet at a satisfactory level. However, these studies generally cover the local genotypes of the country studied [12,13,14]. Our study has genotypes of different origins from many countries, as well as local genotypes and cultivars.

Hence, the distinguishing and superior feature of this planned study compared to previous research was its comprehensive approach. It encompassed a broad range of coriander varieties from diverse climates, geographies, and topographies, covering nearly all coriander-growing regions globally. This extensive variation was characterized by morphological and agronomic traits and by the biochemical properties of local varieties and genotypes. Additionally, this study incorporated UPOV (International Union for the Protection of New Varieties of Plants) criteria, enhancing the depth and scope of the analysis. Therefore, assessing quality characteristics such as fruit yield, essential oil yield, and linalool content in coriander populations of foreign origin, alongside evaluating the quality of these coriander seeds, will provide significant benefits.

To our knowledge, this study is the first to determine the different origin coriander genotypes collected from overseas based on the morphological and yield values with UPOV criteria, in addition to essential oil, fixed oil, and their compositions in some promising outstanding in terms of efficiency coriander genotypes.

## 2. Materials and Methods

### 2.1. Plant Material

In this study, the fruits of 114 coriander genotypes obtained from the USDA (United States Department of Agriculture) in 2019, 2 cultivars (Arslan and Gürbüz) obtained from Ankara University Faculty of Agriculture, and 3 cultivars (Erbaa, Gamze, and Pelmus) obtained from the Black Sea Agricultural Research Institute were used. Agricultural and some quality characteristics of a total of 119 genotypes were examined in Bolu ecological conditions, Bolu, Türkiye. All of them were grown in the 2019 growing season and, except 3 genotypes (Ames 14364, Ames 19032, and Ames 20047), adapted to Bolu ecological conditions. The list of genotypes and cultivars used in the research is given in Table 1.

### 2.2. Experimental Details

The experiment was established on 30 April 2019 at Bolu Abant İzzet Baysal University, Faculty of Agriculture, Research and Application area, as 6 blocks in a single row, according to the augmented trial design. The experiment was conducted with 1 m distance between blocks, 3.5 m length of the blocks, and a plot size of 24.6 m × 8 m with a spacing of 0.3 m between rows and 0.1 m between plants. There are 24 rows in each block, and each block is 25.2 m^2^. The soil properties of the experimental area were as follows: low in phosphorus (P) (0.50 kg/ha), rich in potassium (K) (1083.1 kg/ha) and organic matter (3.71%), clayey and slightly alkaline (pH = 7.56), medium-lime (1.14%), and low salinity (0.04%). Average climatic data were recorded between 8.3 and 19.5 °C for temperature, 6.3 and 138.6 mm for rainfall, and 70.0 and 80.9% for humidity during the vegetation period of 2019 from April to October [15]. In the experiment, the fruits were sown on 30 April 2019, and sowing was conducted with an excessive number of fruits (approximately 50), enabling thinning to be performed 10–15 days after emergence to achieve the desired plants in each row. With the planting in the experimental area, 60 kg/ha of diammonium phosphate (DAP) and 25 kg/ha of ammonium sulfate (21% N) were applied as base fertilizers; at the beginning of flowering, 25 kg/ha of ammonium sulfate was applied again as fertilizer. After the seedling days, watering was performed every day as much as the plants needed using the drip irrigation method, and weed control was performed by hand every two days. After 10 randomly selected plants for the morphological and yield values were taken from each row in the blocks, the entire row was harvested between 27 July and 30 September 2019.

### 2.3. Essential Oil Content (%v/w)

Essential oil contents were determined volumetrically with a Clevenger apparatus according to the water distillation method in dried fruits at 35 °C. Approximately 20 g of sample from the dried fruit prepared for analysis was weighed. The weighed sample was placed in a glass Clevenger flask. Approximately 200 mL of pure water was added to the sample. It was subjected to hydrodistillation for approximately 4 h. Then, the essential oil sample, which accumulates in the graduated section and creates a phase difference with water, was read, and the result is recorded in ml. Then, based on the weighing amount, the amount of essential oil was calculated as a percentage mL/100 g (%*v*/*w*) based on dry matter.

### 2.4. Essential Oil Components (%v/v)

Essential oil components were prepared in the Central Laboratory of Bolu Abant İzzet Baysal University. Samples were diluted 1:100 with methanol for analysis. Essential oil component analysis of the samples was carried out using a GC/GC-MS (gas chromatography (Thermo Scientific Trace 1300) mass detector (Thermo Scientific ISQ QD, Waltham, MA, USA)) device and a capillary column (TG-624; 30.0 m × 0.25 mm × 1.4 μm). In the analysis, helium was used as the carrier gas at a flow rate of 1.00 mL/min, and the samples were injected into the device at 1 μL. The injector temperature was kept at 220 °C, and the column temperature program was set as 70 °C (2 min), 70 °C to 200 °C at 3 °C/minute, and 200 °C (15 min) in splitless mode. In line with this temperature program, the total analysis time was 60 min. For the mass detector, a scanning range (*m*/*z*) of 40–650 atomic mass units and electron bombardment ionization of 70 eV, a transfer line temperature of 250 °C, and an ion source temperature of 220 °C were used. Data from WILEY and NIST libraries were used to identify the components of the essential oil. MS was used to identify component percentages and components of the results.

### 2.5. Fixed Oil Contents (%v/w)

The fixed oil content (%*v*/*w*) was determined in the hexane (C_6_H_14_) extraction of the samples taken from each row according to the Soxhlet method for 8 hours. Fruits of coriander genotypes, 10 g, were used and extracted with a Soxhlet extractor at 60 °C for 8 h using n-hexane as the solvent. After oil extraction, the solvent was removed with a rotary evaporator.

### 2.6. Fixed Acids (%)

To determine the fixed acids, oil samples were esterified according to the principles given in Anonymous [16]. Esters were injected into the gas chromatograph, and the fixed acids were determined as a percentage. Fixed acids were performed using automatic sampling (Shimadzu-AOC20i) and GC (Shimadzu-2010 Plus) (Shimadzu, Kyoto, Japan) with flame ionization detector (FID) and Rtx-2330 Capillary Column (60 m, 0.25-mm inner diameter, 0.2 μm film thickness). The detector temperature was set at 240 °C. Meanwhile, the oven temperature was kept at 140 °C for 5 min. Afterwards, it was increased by 4 °C every minute, brought up to 260 °C, and kept for 20 min. The sample amount was 1 μL, and the carrier gas He control was ensured at 1 mL/min. Fixed acids were identified by comparing the arrival times of the standard 37-component FAME mixture.

### 2.7. Morphological Characteristics and UPOV Criteria

The morphological characteristics in the study were determined as follows: Days to 50% seedlings, flowering, and fruit setting were recorded as the number of days from sowing until 50% of the plants in a net plot produced seedlings, flowers, and fruits as determined by visual observation, respectively. Plant height was measured in centimeters (cm) for ten randomly selected plants, from ground level to the apex, at the time of physiological maturity in the net plot area. The average number of primary branches that emerged directly from the main shoot was counted for ten randomly selected plants at physiological maturity. The number of umbels from 10 randomly selected plants in each row was counted, and the average number of umbels per plant was calculated. The number of umbelletes from 10 randomly selected plants in each row was counted, and the average number of umbelletes per plant was calculated. Thousand fruit weights were calculated from the portion of fruits separated as pure seed from each row; four samples of 100 fruits were weighed using a precision scale. The average weight from these four measurements was then multiplied by 10 to determine the thousand grain weight in grams. After harvest, the fruits from the threshed plants were weighed, and the fruit yield per plant was recorded in g/plant. Biological yield per plant (g) was calculated on the dry weight of the harvested plants. The harvest index was calculated by dividing the biological yield by the fruit yield and multiplying the result by 100 using the following formula.
Harvest index = (Fruit yield/Biological yield) × 100

UPOV criteria examined in coriander genotypes and cultivars were determined according to UPOV [17]. The examined criteria, notes, and explanations are given in Table 2.

### 2.8. Statistical Analysis

The data obtained in the study were subjected to variance analysis according to the augmented trial design, separately for each property, and the significance tests were made using the AVCI statistical program according to the F test and the difference grouping of the means according to the least significant difference (LSD) method [18]. As a result of variance analysis, among genotypes, the following were calculated: 1- Two corrected applications (lines) in different blocks; 2- Corrected application (line) and control (std) difference; 3- Two corrected applications (lines) in the same block; and 4- Four control (std) variances and LSD values. The least significant difference was found by calculation, according to Peterson [19]. In statistics, letters “A” in uppercase, “z” in lowercase, and numbers were used to denote statistical differences among the examined properties, respectively. In addition, cluster analysis was performed between coriander cultivars and genotypes in terms of morphology, yield, and UPOV criteria using the JMP 14 statistical program, and genetic differences were determined. Heat maps and PCA analyses were conducted based on the fixed oil content, major essential oil compositions, and major fixed acids. Also, correlation analysis was carried out to determine the relationship among the morphological and yield values.

## 3. Results and Discussion

### 3.1. Phenotypic Properties

The results of the growing season indicated that different origin coriander genotypes and cultivars showed statistically significant differences in terms of phenotypic properties at a 5% level. The first 10 genotypes and cultivars that stood out in terms of the days to seedling (50%), 50% flowering, and 50% fruit setting days are given in Table 3.

Days to seedling (50%) differ significantly in all the coriander genotypes, ranging from 24.37 to 40.97 days (Appendix A). The Ames 18573 genotype showed the minimum days, followed by the Ames 18567 and Ames 23627. The Ames 18559 and Ames 18589 genotypes were found to take maximum days to seedling. The cultivars (average 33.37 days) compared to genotypes had the earliest seedling days from 64 genotypes. Previous studies reported that seedling days changed between 19.46 and 20.80 days [20] and 15 and 20 days [21]. The obtained seedling day values were found to be different from the previous studies. The differences can be explained by the different sowing times, genotype differences, climate, and growing conditions. Also, it can be noted that low temperature effected the coriander seedling positively due to promoting the breakdown of existing proteins in fruits to specific amino acids, which are necessary for the growth of the embryo [22].

The 50% flowering days are one of the most important properties to selection for the earliness in plant breeding programs. In this study, coriander genotypes showed wide variations in relation to 50% flowering days and ranged from 52.40 to 72.20 days among the coriander genotypes (Table 3). The early 50% flowering was observed in Ames 18569 and Ames 18561 genotypes, followed by Ames 18596, Ames 18571, Ames 18567, Ames 10234, and Ames 18572 (Table 3). The latest 50% flowering days were found from Ames 24907 and Ames 24909 genotypes. Gökduman [23] reported a range of 51 to 79 days for coriander under Isparta ecological conditions, and Moniruzzaman et al. [24] noted that 50% flowering time of 14 coriander genotypes changed between 57.33 and 134.30 days.

In addition, the 50% fruit setting days of coriander genotypes showed statistical difference and changed between 61.40 and 96.20 days (Appendix A). The earliest fruit setting days were obtained from Ames 18567 and Ames 18561 genotypes, followed by Ames 12778, Ames 18569, Ames 18572, and Ames 18572 genotypes (Table 3). The latest values were found in PI 170319 and Ames 24909 genotypes. When the genotypes were compared with the cultivars (average 74.60 days), it was observed that 59 genotypes set fruit late and 52 genotypes set fruit early. No previous study has been found on the 50% fruit setting days of coriander genotypes. So, this study is the first for the determination of the 50% fruit setting days of coriander.

In fact, a positive association between days to seedling (50%) and days to flowering (50%) is expected. However, in our study, the early seedling days could not show a meaningful and significant relationship with the early flowering days. Differences in early seedling days and early flowering days among coriander genotypes have been attributed to variations in temperature, light intensity, and competition for resources between reproductive and vegetative tissues [25]. In addition, genes for phenology and plant development, their interactions with each other, and the environment may affect the late or early flowering [26]. It was determined that 50% flowering and 50% fruit setting days were found partially similar among the genotypes. The differences can be explained by reducing pollination and fruit setting for insect-pollinated coriander genotypes that flowered earlier [27].

### 3.2. Plant Height and Branch Number

The results revealed that statistical differences were found for the plant heights among the coriander genotypes (Appendix A). The first 10 genotypes and cultivars that stood out in terms of the plant height and branch number values are given in Table 4. The plant height value showed wide variations, and it changed between 25.44 and 84.30 cm. The highest plant height was found in the PI 124179 genotype, followed by Ames 21655, Ames 29172, and Ames 18578 genotypes (Table 4). Plant height values of sixteen coriander genotypes were found higher than coriander cultivars. The lowest plant height values were obtained from Ames 23626, Ames 14363, and PI 274290 genotypes. Compared to the obtained values, there is a 30.18% difference between the highest and the lowest plant heights. Previous studies reported that the plant height of coriander changed between 19.7 and 35.8 cm [14], 33.77–67.86 cm [28], 40.2–69 cm [29], 66.86–84.78 cm [30], and 54.8–64.3 cm [31]. The results of the plant height in this study were in correspondence with the findings of reported previous studies.

Branch number per plant is so important for the breeding of the coriander due to directly impacting fruit yield [11]. Statistically significant differences were found among the coriander genotypes for branch number (Table 4). The branch number per plant showed high variability and ranged from 2.82 to 15.10 numbers. The most branch number was found in the PI 193769 genotype, followed by Ames 13900, Ames 23623, and PI 269472 genotypes. Ames 23626 had the lowest branch number, followed by Ames 23625 and Ames 23627 genotypes. Forty-nine genotypes were found to have higher branch numbers compared to cultivars. The positive relationship was detected between the plant height and branch number values in coriander genotypes. Previous studies reported that branch number of coriander changed between 4.67 and 15.20 number [28] and 5.13–7.33 number [32]. The obtained branch number values in this study (2.82–15.10 number) were found partly similar with previous studies.

### 3.3. Number of Umbels per Plant and Number of Umbellets per Plant

There were statistically significant differences for different coriander genotypes with respect to the number of umbels in the experimental year. The differences between the highest and the lowest values showed that the genotypes contained in the present study are quite various (Appendix A). The first 10 genotypes and cultivars that stood out in terms of the umbel number and umbellet number values are given in Table 4. The greatest number of umbels was obtained from Ames 13900 (72.68), followed by Ames 13899 (62.61), PI 193769 (52.41), and Ames 10235 (43.61), whereas the lowest number of umbels was noted for the PI 172808 (2.81) and PI 170319 (3.61) genotypes. Twenty-eight genotypes had higher umbel numbers compared to cultivars. Arslan had the highest umbel number, and Gamze had the lowest value among the coriander cultivars. In earlier studies, Hongal et al. [32] reported that variability of yield and quality parameters in different coriander genotypes showed a wide range in umbel number per plant (9.37–18.03). Similarly, Devi and Sharangi [33] reported that the umbel number of coriander genotypes under Gangetic alluvial soils ranged from 17.34 to 29.98 per plant. A study conducted by Santha et al. [14] determined the morphological and seed yield traits of coriander local land races of Tamil Nadu and reported the number of umbels per plant changed between 4 and 34.6 under irrigated and rain-fed conditions. The previous studies showed that the number of umbels in coriander can show variability depending on the genotypes, growing and soil conditions, or geographical regions. With these possibilities, our results on the number of umbels were found to be partly similar to the previous studies.

The number of umbellets per plant was changed among the coriander genotypes and cultivars (Appendix A). The existence of a wide difference between the highest and the lowest values was quite diverse in the vegetation period. The different origin genotypes were Ames 13900 (originating from Tajikistan) and PI 193769 (originating from Ethiopia), and the property making them distinct was the highest number of umbellets (380.19 and 315.31, respectively) (Table 4), in contrast with other genotypes showing a number of umbellets in the range of 0.19–200.19 (Appendix A). However, the lowest number of umbellets were found from different origin genotypes, such as Ames 12778, Ames 18567, and Ames 18572, with 0.19 per plant. Eighteen genotypes had a greater number of umbellets per umbel over than all other remaining genotypes, which is considered very high fruit yield. Especially, the Ames 13900 genotype had the highest number of umbellets, and this property directly affected positively the fruit and biological yield values of Ames 13900.

The obtained number of umbellets per plant results (0.19–380.19 number) in this study were compared to previous studies; Joble et al. [34] reported between 141.23 and 239.63 number/umbellet per plant; Bhandari and Gupta [35] noted between 7.38 and 174.31 number/umbellet per plant; and Qureshi et al. [36] found between 121 and 336 number/umbellet per plant. The results were found to be partly similar with previous studies. The differences can be explained by the growing conditions, soil properties, ecological conditions, and genotype differences.

### 3.4. Yield and Yield Attribute Properties

One of the yield attributes of coriander is the thousand fruit weight as a quantitative property. This property can be used for a successful selection breeding program. Among the yield components, 1000 fruit weight seems to be the most important. High yielding ability in genotypes could be attributed to significantly higher thousand fruit weight. Especially, significant differences were found among the coriander genotypes for the thousand fruit weight, and it varied between 1.34 and 21.49 g (Appendix A). The first 10 genotypes and cultivars that stood out in terms of the 1000 fruit weight, fruit yield, biological yield, and harvest index are given in Table 5. The lowest thousand fruit weight was detected in Ames 14363 genotype, and the highest thousand fruit weight was determined in Ames 18569 genotype. When the genotypes were compared with the cultivars (average 10.32 g), it was observed that 38 genotypes had high values and 73 genotypes had low values. While the genotypes with the highest values in terms of thousand fruit weight were Ames 18569 (21.49 g), Ames 18572 (20.74 g), Ames 18585 (19.42 g), Ames 18571 (17.62 g), and Ames 23635 (17.06 g), the genotypes with the lowest values were Ames 14363 (1.34 g), Ames 10235 (3.22 g), Ames 29173 (3.23 g), PI 170320 (4.48 g), and Ames 24923 (4.54 g). Previous studies reported that the 1000 fruit weight of the coriander changed between 8.00 and 12.00 g [24] and between 9.33 and 13.82 g [37]. The obtained results from this study were found to be partly similar with the previous studies.

The fruit yield is a very important property in terms of selecting the high yield genotype for breeding purposes. It also has commercial importance in the spice industry. Among the 114 genotypes and five cultivars tested, the Ames 13900 genotype recorded the highest fruit yield per plant, which was followed by the Ames 10234 and Ames 18573 genotypes (Table 5). The lowest values were recorded in Ames 18567 and Ames 24926 genotypes. Seventeen genotypes had the higher fruit yield compared to coriander cultivars. While the Erbaa cultivar had a higher value, the Pelmus cultivar had the lowest fruit yield value among the coriander cultivars. Devi and Sharangi [29] stated that fruit yield per plant ranged from 1.15 g to 6.17 g. It was reported that the precipitation and temperatures impact the yield and yield product of the coriander [38]. Moreover, different sowing times of coriander decreased the fruit yield between 30.6 and 76.4% [39]. Delaying in sowing time decreased the both fruit yield and biomass yield of coriander by 76.4 and 74.7, respectively [40]. In addition, many more studies reported that productivity of coriander decreased as sowing was postponed to the latest dates [41,42,43].

There were statistically significant differences in coriander genotypes for the biological yield, and it changed between 0.01 and 50.78 g per plant (Appendix A). The Ames 13900 and PI 193769 genotypes demonstrated the highest biological yield values compared to other genotypes (Table 5). The lowest biological yield values were observed in Ames 12778, Ames 14363, and Ames 18567 genotypes lower than 1 g. Generally, seventeen coriander genotypes had higher biological yield compared to cultivars, and the Erbaa cultivar had the highest biological yield among the cultivars. Rashed and Darwesh [44] reported that a significant positive relationship was observed between the biological yield and average minimum temperature in the vegetation period of coriander, and the biological yield increased with the increasing of the average minimum temperature. Thakur and Thakur [45] reported that the biological yield of the coriander changed between 31.54 and 60.61 g/plant under growth water stress conditions. The obtained results were found to be partly similar to the findings of Thakur and Thakur [45]. In this study, it is thought that high average rainfall (69.4 kg/m^2^), high humidity (73.1%), and low temperature (16.1 °C) values cause a decrease in the biological yield of coriander genotypes.

Statistically significant differences were found among the coriander genotypes for the harvest index (Appendix A). The harvest index ranged from 7.79% to 73.36%, and Ames 23640, Ames 24921, and Ames 18595 genotypes had the highest values (Table 5 and Appendix A). The PI 669964 and PI 193769 genotypes showed the lowest values for harvest index. Thirty-two coriander genotypes were found to have lower values compared to coriander cultivars. The Gürbüz cultivar had the highest value, and the Pelmus cultivar had the lowest value among the coriander cultivars. In earlier studies, Nagappa et al. [46] reported that variability of combined analysis results on harvest index values of thirty coriander genotypes showed a wide range between 21.02 and 57.75%, and Kassu et al. [47] reported that the coriander showed variability harvest index between 18.90 and 42.10% depending on the different sowing dates. The present results comply with the mentioned results with respect to harvest index.

### 3.5. Essential Oil and Essential Oil Components

The secondary metabolites of the plants may change in quality and quantity based on the genotype, tissue, growth period, geographical area, and stress factors as abiotic and biotic factors [48]. For the efficacy and the informative value of the released essential oils of coriander fruits, the composition of terpenes is crucial, influenced by the different origin genotypes and growing conditions. In total, essential oil contents were determined in 91 coriander genotypes (86 coriander genotypes and five cultivars) selected as promising based on the high yield and other properties according to the sufficient fruit quantity to obtain essential oil contents (Table 6). Similarly, essential oil compositions and fixed acids were determined in 40 coriander genotypes (35 coriander genotypes and five cultivars) selected as promising on the basis of the high yield and, most importantly, other properties, such as umbels/plant, branches/plant, 1000 fruit weight, early flowering, and early fruit setting days, which directly or indirectly affected fruit yield (Table 3, Table 4 and Table 5).

Essential oil contents (EOCs) of the coriander genotypes were isolated in 86 genotypes and five cultivars depending on the fruit yield values. EOCs of the coriander genotypes ranged from 0.05%*v*/*w* to 1.86%*v*/*w*. The highest EOC was found in Ames 13900, followed by PI 502320 and PI 531293 genotypes. The lowest EOC were observed from Ames 23624, Ames 23621, PI 269472, and PI 193493 genotypes. In general, 30 genotypes were found to be higher than all cultivars in terms of essential oil content, while five genotypes were found to be lower than all cultivars. It was determined that more than 50% of the genotypes whose essential oil contents were determined due to their fruit yield values had essential oil contents above 0.50%*v*/*w*.

The essential oil content of the coriander showed high different variabilities depending on the country, region, growing conditions, or other factors. Ebrahimi et al. [12] conducted a study to determine the essential oil compositions of 19 coriander accessions in Iran, and it was noted that the essential oil content of the dried seeds changed between 0.1 and 0.36%*v*/*w*. Fattahi et al. [49] found the essential oil content of dried shoot sample between 0.18 and 0.31%*v*/*w* under cadmium and lead stress conditions.

The international standard for coriander essential oil varies by country. According to the European Pharmacopoeia, the essential oil value of coriander should not be less than 0.2%*v*/*w* in Indian origin, 0.8–1%*v*/*w* and 0.5%*v*/*w* in Russian origin. In addition, according to the Turkish Food Codex Spice Communiqué, the essential oil content of unground coriander spice is required to be not less than 0.4%*v*/*w* [29]. The result of the study revealed that five genotypes and Arslan cultivars had lower than 0.2% essential oil contents, and these genotypes were found to have lower values than European Pharmacopoeia standards.

In total, forty-three essential oil compositions were detected in different coriander genotypes. Some differences in the quantity of the main components of essential oils extracted from fruit of the different origin coriander genotypes and cultivars. The total oil compositions of fruits were found to be in the range of 77.23–93.85%*v*/*v*, with linalool, camphor, ɣ-terpinene, *p*-cymene, and β-pinene being the most abundant compounds, constituting around 18.94 to 71.13%*v*/*v* of the investigated total concentration of essential oils (Table 7). The first major essential oil composition was linalool, whose content varied from 3.13 to 45.70%*v*/*v*. It was observed that 11 genotypes had high values and 24 genotypes had low values compared to cultivar means (13.03%). The genotypes with the highest values in terms of linalool ratio were Ames 29174 (45.70%*v*/*v*), Ames 23635 (35.00%*v*/*v*), Ames 18585 (33.87%*v*/*v*), Ames 24907 (30.25%*v*/*v*), and Ames 24921 (25.81%*v*/*v*) genotypes. The genotypes with the lowest values were Ames 18577 (3.13%*v*/*v*), Ames 18590 (5.56%*v*/*v*), Ames 18573 (5.73%*v*/*v*), Ames 18591 (6.08%*v*/*v*), and Ames 18569 (6.29%*v*/*v*) genotypes. The second major essential oil composition was ɣ-terpinene. The values of the ɣ-terpinene ranged from 0.05 to 15.60%*v*/*v*. Ames 29174 and Ames 24907 genotypes had the lowest and highest ɣ-terpinene values, respectively. The ɣ-terpinene means of the cultivars were found higher than 17 coriander genotypes. The third major essential oil composition was camphor, which changed between 0.05 and 11.18%*v*/*v* (Table 7). The highest camphor contents were found in the Pelmus cultivar, followed by the Ames 33640 genotype. The lowest value was noted in the Ames 20048 genotype. It was noted that seven genotypes had high values and 28 genotypes had low values compared to cultivars means (5.51%*v*/*v*). *p*-cymene was the fourth essential oil composition of the coriander genotypes, and it changed between 0.02 and 6.90%*v*/*v*. The lowest *p*-cymene ratio was detected in Ames 29174 genotype, and the highest *p*-cymene ratio was detected in Ames 23634 genotype. When genotypes were compared with cultivars (average 4.02%), it was observed that 19 genotypes had high values and 16 genotypes had low values. The last major essential oil composition was noted as β-Pinene. It ranged from 0.10 to 2.68%, and Ames 18587 and Ames 10235 genotypes showed the lowest and highest β-Pinene contents, respectively. When the genotypes were compared with the cultivars (0.69% on average), it was noted that 24 genotypes had high values and 11 genotypes had low values. The genotypes with the highest β-pinene content were Ames 10235 (2.68%*v*/*v*), Ames 29174 (2.07%*v*/*v*), PI 193493 (1.94%*v*/*v*), PI 478378 (1.94%*v*/*v*), and Ames 18590 (1.86%*v*/*v*); the lowest genotypes are Ames 18587 (0.10%*v*/*v*), Ames 10234 (0.19%*v*/*v*), Ames 23635 (0.36%*v*/*v*), Ames 24921 (0.37%*v*/*v*), and Ames 18568 (0.39%*v*/*v*).

In this study, other minor essential oil compositions (38) i.e., a-pinene, camphene, sabiene, di-limonene, and 1-borneol were found lower than 10%*v*/*v* contents (Appendix A). These minor essential oil compositions showed high variability among the coriander genotypes.

Many previous studies reported that the main essential oil compositions of the coriander fruits were linalool, *p*-cymene, and limonene. However, the essential oil compositions may show variation based on the different soil and climatic conditions, altitude, seasonal factors, another environmental effect, and different chemical variations or chemotypes can be revealed in some cases [50].

Mandal and Mandal [51] reported that linalool is the major and greatest (40–70%*v*/*v*) essential oil composition of coriander fruits. This component is found in many plants as a monoterpene alcohol and shows an antimicrobial effect on the different bacteria. The result in this study showed that linalool content was found to be partly similar with the Mandal and Mandal [51].

The main essential oil compositions may vary depending on the genotype, geographical conditions, agricultural applications, climatic, and soil properties. Weisany et al. [52] reported that coriander apiole and carvone were the first two main essential oil compositions in coriander/soybean intercropping and mycorrhizae application. Another study reported that n-decanal, 2E-dodecanal, 2E-decanal, 2E-tridecen-1-al, and n-nonane were the main among the 24 essential oil compositions of coriander leaves grown under salinity and foliar-applied silicon [53]. Similarly, El-Zaeddi et al. [54] reported that the main compositions were noted as E-2-dodecenal, dodecanal, and octane at different harvest times of coriander in Spain ecological conditions.

The differences in essential oil contents and compositions can be explained by some factors such as growing conditions (irrigation, used fertilizer species and doses), environmental conditions (light, nutrient availability, temperature, and day length), maturation process, or different genotypes [7,8,55,56,57,58]. Sangwan et al. [59] reported that the linalool contents of coriander fruits were affected by climatic factors such as cloudy days, lower temperatures during maturation, and high rainfall amounts during the vegetation periods.

### 3.6. Fixed Oil Content and Fatty Acid Profiles

The results showed that statistically significant differences were found for the fruit-fixed oil of the coriander genotypes (Table 8). The fixed oil content varied between 10.22 and 34.03%*v*/*w*. When genotypes were compared with cultivars (average 28.24%*v*/*w*), it was observed that 13 genotypes had high values and 22 genotypes had low values. The genotypes with the highest fixed oil contents were Ames 10234 (34.03%*v*/*w*), Ames 23634 (33.63%*v*/*w*), Ames 18559 (33.19%*v*/*w*), Ames 13900 (32.21%*v*/*w*), and Ames 33640 (31.85%*v*/*w*), while the genotypes with the lowest genotypes were Ames 10235 (10.22%*v*/*w*), Ames 18568 (10.35%*v*/*w*), Ames 18587 (12.52%*v*/*w*), Ames 18585 (13.56%*v*/*w*), and Ames 18575 (13.72%*v*/*w*).

Previous studies reported that fixed oil of coriander changed between 5.8 and 22.9%*v*/*w* [59], 1.63–24.26%*v*/*w* [29], 15.01–17%*v*/*w* [60] and 4.6–25.1%*v*/*w* [61]. The obtained results from this study on fixed oil were found to be similar with the previous studies.

In the fruits of different coriander genotypes and cultivars, eighteen fixed oil acids were identified (85.10–100.00%*v*/*v* of total oil samples), with petroselinic, palmitic, elaidic, behenic, and arachidic acids being the most abundant compositions that totally constituted around 62.62–99.48%*v*/*v* of the investigated fixed acids (Table 9 and Appendix A). Petroselinic asit (C18:1n12) was found in the highest quantity (24.47–87.70%*v*/*v*), followed by elaidic acid (C18:1n9t) (1.55–47.44%*v*/*v*), palmitic acid (C16:0) (7.13–23.04%*v*/*v*), behenic acid (C22:0) (3.17–12.56%*v*/*v*), and arachidic acid (C20:0) (1.08–13.33%*v*/*v*).

The variability in the content of petroselinic acid was found to be highly different among the coriander genotypes, and it ranged from 24.47 to 87.70%*v*/*v* (Table 9). The highest petroselinic acid content was obtained from the Ames 18559 genotype with 87.70%, followed by the Ames 13900 with 80.53%*v*/*v* and Ames 13899 with 79.71%*v*/*v*. The lowest petroselinic acid was obtained from Gamze cultivar with 24.47%*v*/*v*, and followed by Ames 18587 genotype with 31.17%*v*/*v*. Comparing cultivars, it was observed that the cultivars of Gürbüz (71.65%*v*/*v*) and Arslan (65.52%*v*/*v*) had higher petroselinic acid values than 25 genotypes and other cultivars. The obtained results on the petroselinic acid values of different origin coriander genotypes from this study were found to be similar to those of Sriti et al. [60], Nguyen et al. [61], and Nguyen et al. [62], who reported the petroselinic acid values between 42.20 and 76.37%*v*/*v*, 48.80 and 76.40%*v*/*v* and 51.80 and 74.00%*v*/*v*, respectively.

The elaidic acid values of the coriander genotypes showed wide variation, and these values changed between 1.55 and 47.44%*v*/*v* (Table 9). The Gamze cultivar, with PI 478378 and PI 174129 genotypes, had the highest values, and Ames 18569, Ames 24907, and Ames 18573 genotypes had the lowest elaidic values. This study could be the first report on the elaidic acid of coriander because no previous study has been found on the elaidic acid ratios of coriander.

In the study, palmitic acid values of coriander genotypes were found to be statistically significant at the 0.05% level (*p* < 0.05) (Table 9). As seen in Table 9, palmitic acid values of coriander genotypes varied between 7.13 and 23.04%*v*/*v*. The lowest palmitic acid value was detected in the Ames 18569 genotype, and the highest palmitic acid value was detected in the Gamze cultivar, followed by the Ames 18581 genotype. Three genotypes had high values, and 32 genotypes had low values compared to means of coriander cultivars (14.67%*v*/*v*). Compared to palmitic acid values of coriander genotypes with the results of previous studies, Ertas et al. [63] and Neffati and Marzouk [64] reported high values from this study. However, Nguyen et al. [61] and Nguyen et al. [62] reported similar findings with this study. It could be thought that the differences from this study reported by Ertas et al. [63] and Neffati and Marzouk [64] were due to climate and growing conditions, environmental factors of the region where the research was conducted, genotype differences, and the large number.

As seen in Table 9, the behenic acid values of coriander genotypes varied between 3.17 and 12.56%*v*/*v*. The lowest behenic acid was detected in Ames 18581 genotype, and the highest behenic acid ratio was detected in Ames 33640 genotype. When the genotypes were compared with the cultivars (average 6.25%*v*/*v*), it was observed that 18 genotypes had high values, and 17 genotypes had low values in terms of behenic acid values. The genotypes containing the highest behenic acid values were Ames 33640 (12.56%*v*/*v*), Ames 18587 (10.70%*v*/*v*), Ames 10235 (10.31%*v*/*v*), Ames 23634 (10.19%*v*/*v*), and Ames 24907 (9.24%*v*/*v*), and the lowest genotypes were Ames 18581 (3.17%*v*/*v*), Ames 29174 (3.89%*v*/*v*), Ames 23635 (4.13%*v*/*v*), Ames 18585 (4.15%*v*/*v*), and Ames 18595 (4.19%*v*/*v*). The behenic acid values of coriander genotypes were compared with the results of previous studies; the findings reported by Nguyen et al. [61] (0.01–1.60%*v*/*v*) and Nguyen et al. [62] (0.10–2.30%*v*/*v*) were found lower than the obtained from this study. The higher values in this study may be due to genotypic differences, climatic conditions, sample preparation for acids, and applied analytical procedures and crop management practices.

Arachidic acid values ranged from 1.08 to 13.33%*v*/*v* in all the genotypes (Table 9). The genotype Ames 18566 showed the maximum arachidic acid, followed by Ames 23634 (10.84%*v*/*v*), Ames 18573 (7.94%*v*/*v*), PI 193493 (5.65%*v*/*v*), and Ames 19089 (5.63%*v*/*v*). Ames 33640 had a minimum value for the arachidic acid. Genotypes compared with the cultivars means (5.38%*v*/*v*), it was observed that five genotypes had high values and 30 genotypes had low values.

In earlier studies, Nguyen et al. [61] reported that arachidic acid was one of the major acids, and it changed between 0.10 and 0.30%*v*/*v* in ecological conditions in France. Similarly, Nguyen et al. [62] noted that it was changed between 0.10 and 5.90%, and Sriti et al. [60] reported between 0.15 and 2.19%*v*/*v*. The present results are incompatible with the mentioned results with respect to arachidic acid. Yaldiz and Camlica [7] (2019) and Camlica and Yaldiz [65] reported that the fixed acids vary according to variety, soil structure, and climatic conditions.

The thirteen minor fixed oil acids were found lower than 10%*v*/*v*, and values are given in Appendix A.

### 3.7. UPOV Criteria

Coriander cultivars and genotypes were identified in terms of UPOV criteria (Appendix A). Anthocyanin coloration in the flower was found to be absent (1) in one cultivar (Pelmus) with 6 genotypes and present (9) in four cultivars (Erbaa, Gamze, Arslan, Gürbüz) with 105 genotypes. The intensity of anthocyanin coloration in the flowers was found to be weak (3) in four cultivars (Gamze, Arslan, Gürbüz, and Pelmus) with 81 genotypes, medium (5) in one cultivar (Erbaa) with 28 genotypes, and strong (7) in 2 genotypes. Anthocyanin coloration of hypocotyl in seedlings is absent or very weak (1) in 9 genotypes, weak (3) in 52 genotypes and one cultivar (Arslan), medium (5) in 32 genotypes and four cultivars (Erbaa, Gamze, Pelmus, Gürbüz), and strong (7) in 18 genotypes. The cotyledon shape was found as narrow elliptic (1) in 36 genotypes, elliptic (2) in 43 genotypes and three cultivars (Erbaa, Arslan, Gürbüz), and broad elliptic (3) in 32 genotypes and two cultivars (Gamze, Pelmus). Density of foliage in plants was found to be sparse (3) in 9 genotypes, medium (5) in 72 genotypes and four cultivars (Erbaa, Gamze, Pelmus, Gürbüz), and dense (7) in 30 genotypes and one cultivar (Arslan). Coloration in foliage was found as yellowish green (1) in four cultivars (Erbaa, Gamze, Pelmus, Gürbüz) with 35 genotypes, green (3) in one cultivar (Arslan) with 46 genotypes, and dark green (5) in 30 genotypes. The number of leaflets in basal leaf was found to be three leaves (1) in four cultivars (Erbaa, Gamze, Arslan, Gürbüz) with 54 genotypes and five leaves (2) in one cultivar (Pelmus) with 57 genotypes. The size of the terminal leaflet in the leaf was found to be small (3) in one cultivar (Arslan) with 35 genotypes, medium (5) in two cultivars (Gamze, Gürbüz) with 42 genotypes, and large (7) in two cultivars (Erbaa, Pelmus) with 34 genotypes. Structure of feathering in basal leaf was found to be fine (1) in three cultivars (Arslan, Pelmus, Gürbüz) with 84 genotypes and medium (2) in two cultivars (Erbaa, Gamze) with 27 genotypes. The leaf serration status was found to be sparse (3) in 39 genotypes, medium (5) in 45 genotypes and three cultivars (Erbaa, Arslan, Gürbüz), and dense (7) in 27 genotypes and two cultivars (Gamze, Pelmus). The fruit shape was found to be rounded (2) in five cultivars with 111 genotypes. The brown color intensity in the fruit was found to be light (3) in one cultivar (Pelmus) with 15 genotypes, medium (5) in one cultivar (Arslan) with 58 genotypes, and dark (7) in three cultivars (Erbaa, Gamze, Gürbüz) with 41 genotypes. Erbaa, Gamze, and Gürbüz cultivars were found similar in terms of anthocyanin density in the flower, anthocyanin density difference in the flower, green color intensity in the plant, green color of the plant, number of leaves, and brown color intensity in the fruit. So, the results of the UPOV criteria values showed large variability among the coriander cultivars and genotypes.

### 3.8. Cluster Analysis

Cluster analysis of the coriander cultivars and genotypes used in the study was performed according to morphological, yield, and some UPOV criteria (Figure 1). Also, the cluster analysis was conducted to determine genetic diversity depending on the fixed oil content, major essential oil compositions, and major fixed acids among the high-yielding selected coriander genotypes and cultivars (Figure 2). As a result of cluster analysis, coriander cultivars and genotypes were divided into two main groups (A and B) (Figure 1).

Sixty-six genotypes and two cultivars (Erbaa and Gamze) took place in group A, and 45 genotypes and three cultivars (Arslan, Pelmus, and Gürbüz) took place in group B. Group A is divided into two subgroups, A1 and A2. There were 59 genotypes and two cultivars (Erbaa and Gamze) in subgroup A1. Among the 59 genotypes in this group, 50 genotypes were determined to be Ames-coded genotypes, and 9 genotypes were determined to be PI-coded genotypes. There were seven genotypes in the A2 subgroup. These nine genotypes consist of genotypes originating from Israel, the Netherlands, Germany, the USA, Tajikistan (2), and Ethiopia.

Group B was divided into two subgroups, B1 and B2. There were six genotypes and one cultivar (Pelmus) in subgroup B1. Six genotypes in this group consist of the Ames-coded genotype, and two genotypes were originating from Russia. There were 39 genotypes and two cultivars (Gürbüz, Arslan) in the B2 subgroup. Among the 39 genotypes in this group, 24 genotypes were determined to be Ames-coded genotypes, and 15 genotypes were determined to be PI-coded genotypes. The Ankara-origin coriander cultivars Arslan and Gürbüz were grouped together within the same subgroup of the main category based on the examined properties.

Figure 2 showed that cluster analysis divided into two A and B main groups for coriander genotypes and cultivars based on the fixed oil, major essential oil compositions, and major fixed acids. While most of the coriander genotypes took place in the A main group, three cultivars (Erbaa, Gürbüz, and Gamze) were found in the B main group. These main groups are also divided into two subgroups: A1, A2, B1, and B2. The subgroup A1 contained nine genotypes, and subgroup A2 contained 11 genotypes and 2 cultivars (Arslan and Pelmus). The subgroup B1 included only the Gamze cultivar, and the subgroup B2 included 15 genotypes and two cultivars (Erbaa and Gürbüz). The B1 subgroup was separated with the values of the fixed oil acids as palmitic, elaidic, and petroselinic. The A1 subgroup was separated from the other groups with β-Pinene essential oil composition and behenic acid.

Gamze and Erbaa with Arslan and Pelmus cultivars were found in the same main group in both Figure 1 and Figure 2. As a result, it can be concluded that these cultivars showed similar genetic variation depending on the examined properties.

### 3.9. Heat Map and Principal Coordinate Analyses

The heat map and principal coordinate analyses were conducted to determine the relationship among the coriander genotypes and cultivars with the major essential oil compositions, fixed oil, and major fixed acids (Figure 3 and Figure 4). The heat map analysis revealed that the high-yielding coriander genotypes and cultivars were separated with the fixed oil petroselinic acid values. Especially, the Gamze cultivar took place alone in a group because it included the highest elaidic acid and the lowest petroselinic acid. The Ames 18566 (originating from the Czech Republic) and PI 269472 (originating from Pakistan) showed differences for the arachidic acid and γ-terpinene, respectively.

From five principal components, PC 1 to PC5, extracted from the original data and having an eigenvalue greater than one, accounting for 72.94% of the total variation, suggesting that these principal component scores might be used to summarize the original 11 variables in any further data analysis (Figure 4). Accordingly, the first principal component had high positive coefficients for petroselinic acid. The major contributing character for the diversity in the second principal component (PC2) was linalool, while the third principal component (PC3) was fixed oil content. The fourth principal component contributed diversity for the character of arachidic acid. The fifth principal component was palmitic and behenic acid. On the basis of PCA, most of the important quality properties for the selected genotypes were presented in PC1, PC2, PC3, and PC4. A high PC score for a particular genotype in a particular component showed high values for the variables in that particular genotype.

### 3.10. Correlation Analysis

The correlation analysis was conducted to determine the relationship among the morphological and yield values. A total of 29 correlations were identified among the examined characteristics, with 22 showing highly significant positive correlations (Table 10). The days to 50% fruit setting had the most correlations, interacting with seven of the characteristics studied. The highest significant positive correlation was found between the number of umbels and the number of umbelletes, with a correlation coefficient of r = 0.902 **. This was followed by the correlation between fruit yield and biological yield, which had a coefficient of r = 0.856 **. It was clearly reported that the highly significant positive correlation was found between the flowering and fruit setting days. Positive correlations were also observed between the days to 50% flowering and branch number, as well as between thousand fruit weight and harvest index. Highly significant negative correlations were noted between thousand fruit weight and days to 50% fruit setting, plant height, umbel number, and umbelletes. Additionally, two positive correlations were found: one between days to 50% flowering and branch number, and another between thousand fruit yield and harvest index. A negative correlation was detected between branch number and thousand fruit yield, with a coefficient of r = −0.207 *.

Kumar et al. [66] reported that positive correlations were found between the plant height-branch number, 50% flowering days-branch number, and umbel-umbellet. Similarly, Nagappa et al. [67] noted that the correlation study revealed that plant height, number of primary branches per plant, days to 50% flowering, number of umbels per plant, number of umbelettes per umbel, harvest index, thousand fruit weight, and oil content all exhibited significant positive associations with fruit yield per plant at both the phenotypic and genotypic levels. The obtained results from this study on the correlation analysis showed similar findings with the previous studies.

## 4. Conclusions

The ANOVA analysis reveals that selectable, wide-promising genotypes from the existing genetic stock have desired properties, such as fruit yield, essential oil content, fixed oil ratio, and their compositions and acids. Also, the results of the study showed that the genotypes, which had high yield and yield components with some chemical properties, can be used for coriander production purposes for future breeding studies. So, Ames 13900 and Ames 18595 genotypes can be recommended for the fruit yield, fixed oil, and essential oil contents. Also, these genotypes can be suggested for the petroselinic acid production. The cluster analysis divided into two main groups, and most of the genotypes took place in the A main group. Similar findings were found from the heat map analysis. The PCA analysis showed over 70% variation, and three fixed oil acids, fixed oil content, and linalool essential oil composition were found as major contributors. The morphological, yield, and quality properties of the different origin plants would be significant to select for future breeding programs. So, this study can contribute to the selection of promising coriander genotypes based on the desired properties for the breeders. In conclusion, the promising coriander genotypes may be used for different programs, increasing their areas of use.

## Figures and Tables

**Figure 1 biology-13-00866-f001:**
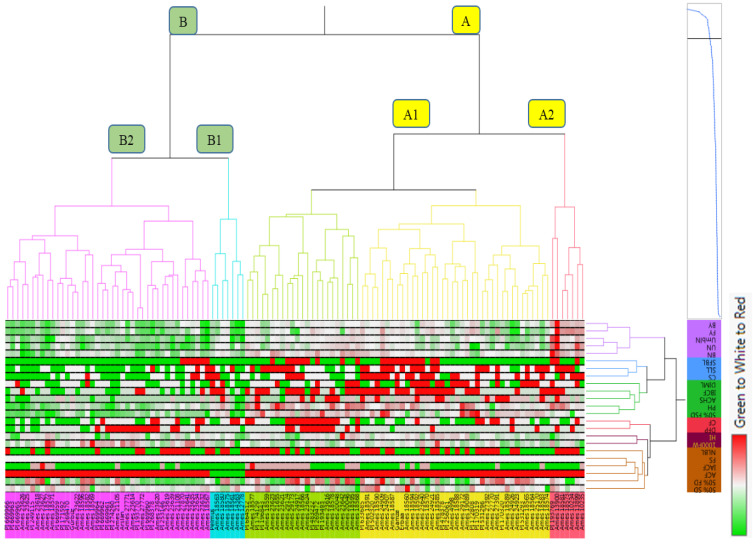
Cluster analysis result of the coriander genotypes depending on the examined morphological and yield values. Green to white to red colors show the difference among the examined properties and genotypes. Green color indicates lower data values, white color shows average value, and red represents higher data values.

**Figure 2 biology-13-00866-f002:**
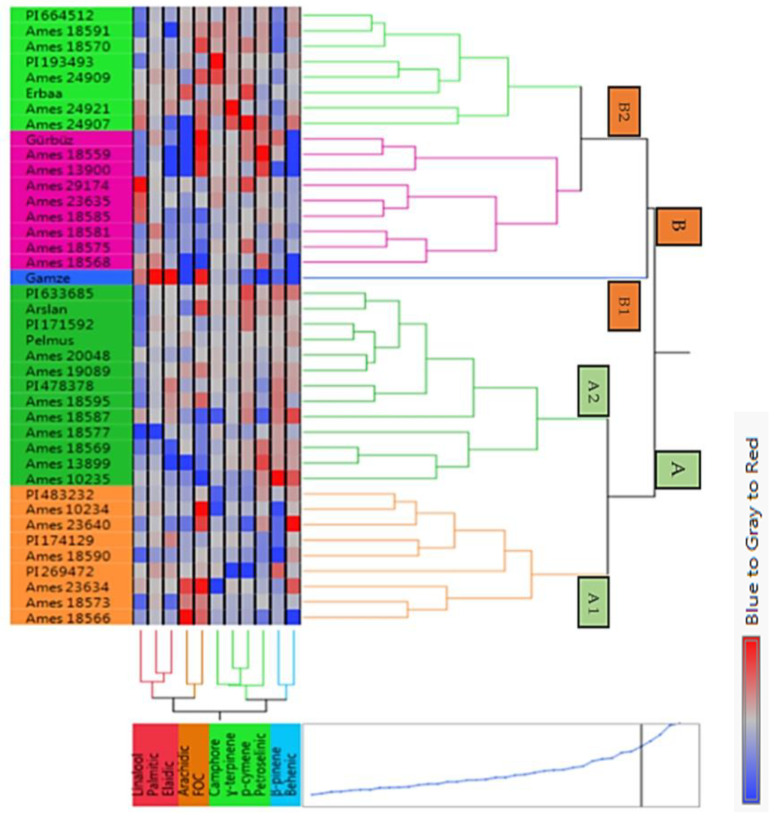
Cluster analysis result of the different origin high-yielding coriander genotypes depending on the examined properties. Blue to grey to red colors show the difference among the examined properties and genotypes. Red and light shades of red colors indicate higher data values, and light shades of blue to blue colors represent lower data values.

**Figure 3 biology-13-00866-f003:**
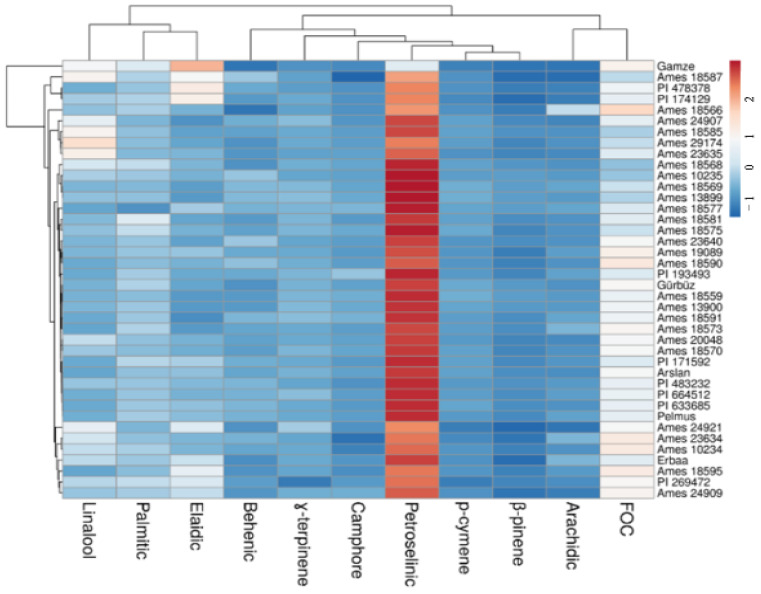
Heat map analysis result of the selected coriander genotypes depending on the examined properties. From orange to red (between 1 and 2) colors indicate higher data values, and from light shades of blue (between 0 and 1) to blue (between −1 and 0) colors represent lower data values.

**Figure 4 biology-13-00866-f004:**
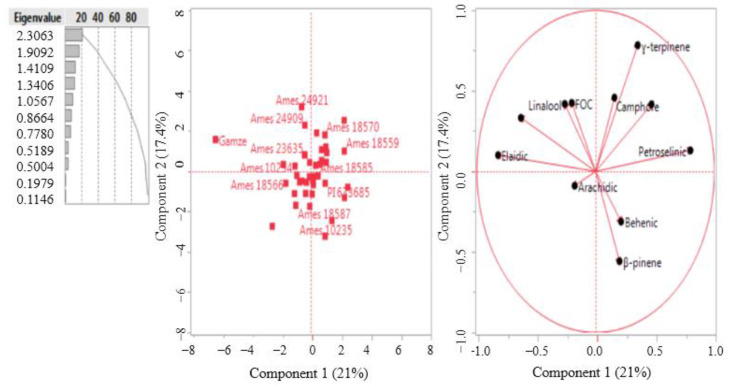
Principal coordinate analysis result of the examined properties in different selected coriander genotypes.

**Table 1 biology-13-00866-t001:** The general information on coriander genotypes and cultivars used in the study.

No	ID	Plant Name	Collected Place	No	ID	Plant Name	Collected Place
1	Ames 4998	896	Türkiye	61	Ames 23632	CORI 88	Oman
2	Ames 10234	Ames 10234	Israel	62	Ames 23634	CORI 90	Oman
3	Ames 10235	Ames 10235	Minnesota, USA	63	Ames 23635	CORI 91	Oman
4	Ames 12778	2410	Nepal	64	Ames 23639	CORI 95	Oman
5	Ames 13899	USSR 90-08-07	Tajikistan	65	Ames 23640	CORI 96	Oman
6	Ames 13900	USSR 90-08-08	Tajikistan	66	Ames 23641	CORI 97	Oman
7	Ames 14363	Ames 14363	Egypt	67	Ames 23642	CORI 133	Pakistan
8	Ames 14364	Ames 14364	Egypt	68	Ames 24907	ISN 187	Bulgaria
9	Ames 18559	880558	N. Rhine-Westphalia, Germany	69	Ames 24909	Coriander Long Standing	USA
10	Ames 18560	880566	N. Rhine-Westphalia, Germany	70	Ames 24917	A008	Portugal
11	Ames 18561	880589	Quebec, Canada	71	Ames 24921	A012	USA
12	Ames 18563	880780	Calvados, France	72	Ames 24923	A014	Georgia
13	Ames 18564	880850	Maine-et-Loire, France	73	Ames 24926	A017	Azerbaijan
14	Ames 18565	880885	Berlin, Germany	74	Ames 25170	CO94INC 1121	USA
15	Ames 18566	880886	South Moravia, Czech Republic	75	Ames 27391	Z023	Uzbekistan
16	Ames 18567	880972	Madrid, Spain	76	Ames 27392	Z075	Uzbekistan
17	Ames 18568	883197	Tuscany, Italy	77	Ames 27770	CO SA 3-02	Palestine
18	Ames 18569	883216	Ibaraki, Japan	78	Ames 27771	CO SA 3-03	Palestine
19	Ames 18570	883217	Ibaraki, Japan	79	Ames 27772	CO SA 3-04	Palestine
20	Ames 18571	883218	Ibaraki, Japan	80	Ames 27870	CO SA 3-05	Palestine
21	Ames 18572	883258	North Holland, Netherlands	81	Ames 29172	GSMO 1-2	Georgia
22	Ames 18573	883260	Gelderland, Netherlands	82	Ames 29173	GSMO 2-8	Georgia
23	Ames 18574	883261	Central Bohemia, Czech Republic	83	Ames 29174	GSMO 8-3	Georgia
24	Ames 18575	883296	Leningrad, Russia	84	PI 170319	No. 2348	Türkiye
25	Ames 18577	883297	Leningrad, Russia	85	PI 170320		Türkiye
26	Ames 18578	883298	Leningrad, Russia	86	PI 171592	TU2_Ic	Türkiye
27	Ames 18580	883300	Leningrad, Russia	87	PI 172808	Maydonoz Gili; Kinzi	Türkiye
28	Ames 18581	891044	Germany	88	PI 174129	TU4_IIIb	Türkiye
29	Ames 18582	891121	Arhus, Denmark	89	PI 193493		Ethiopia
30	Ames 18583	891124	Seine-Maritime, France	90	PI 193769	Dimbelal	Ethiopia
31	Ames 18585	891139	South Moravia, Czech Republic	91	PI 193770	Dimbelal	Ethiopia
32	Ames 18587	891161	Gelderland, Netherlands	92	PI 196843		Ethiopia
33	Ames 18588	891280	Warszawa, Poland	93	PI 249115		India
34	Ames 18589	891281	Poland	94	PI 253146	Gashniz	Iran
35	Ames 18590	891282	Poland	95	PI 256061	AF1_IIIb	Afghanistan
36	Ames 18591	891358	Saxony-Anhalt, Germany	96	PI 268378	Gashniz	Afghanistan
37	Ames 18592	901014	England	97	PI 269470		Pakistan
38	Ames 18594	901016	England	98	PI 269472		Pakistan
39	Ames 18595	901050	Romany	99	PI 274290	1180	India
40	Ames 18596	901069	Canada	100	PI 478378	O 21	China
41	Ames 19032	Ames 19032	Kazakhstan	101	PI 483232	CHL1_IIIb	Chile
42	Ames 19089	Ames 19089	Kazakhstan	102	PI 502320	AR-235	Uzbekistan
43	Ames 20046	VIR 79	Azerbaijan	103	PI 531293	CSILLAG	Hungary
44	Ames 20047	VIR 97	Armenia	104	PI 531296	441	Hungary
45	Ames 20048	VIR 101	Kazakhstan	105	PI 633685	CDC Major	Canada
46	Ames 21105	IC 67145	India	106	PI 664512	TJK 2006:074	Tajikistan
47	Ames 21108	IC 67155	India	107	PI 669959	IC 33013	India
48	Ames 21655	G18171	Russia	108	PI 669960	IC 33667	India
49	Ames 23614	CORI 189	Saarland, Germany	109	PI 669961	IC 34513	India
50	Ames 23616	CORI 220	Russia	110	PI 669962	IC 33728	India
51	Ames 23618	CORI 115	Sudan	111	PI 669963	IC 62485	India
52	Ames 23619	CORI 127	Bhutan	112	PI 669964	IC 67148	India
53	Ames 23620	CORI 131	Baluchistan, Pakistan	113	PI 669965	IC 67153	India
54	Ames 23621	CORI 82	Syria	114	PI 669966	IC 67161	India
55	Ames 23622	CORI 84	Syria	115	Arslan	Cultivar	Türkiye
56	Ames 23623	CORI 86	Syria	116	Erbaa	Cultivar	Türkiye
57	Ames 23624	CORI 87	Syria	117	Gürbüz	Cultivar	Türkiye
58	Ames 23625	CORI 98	China	118	Pelmus	Cultivar	Türkiye
59	Ames 23626	CORI 117	Sudan	119	Gamze	Cultivar	Türkiye
60	Ames 23627	CORI 119	Sudan				

**Table 2 biology-13-00866-t002:** UPOV criteria of the different origin coriander genotypes.

No	Criteria	Note	Explanation	No	Criteria	Note	Explanation
1	Anthocyanin coloration in the flower	1	Absent	7	Number of leaflets in basal leaf	1	Three
		9	Present			2	Five
2	Intensity of anthocyanin coloration in the flowers	3	Weak	8	Size of terminal leaflet in leaf	3	Small
		5	Medium			5	Medium
		7	Strong			7	Large
3	Anthocyanin coloration of hypocotyl in seedling	1	Absent/very weak	9	Structure of feathering in basal leaf	1	Fine
		3	Weak			2	Medium
		5	Medium			3	Coarse
		7	Strong	10	Density of incisions on margin in leaflet	3	Sparse
		9	Very strong			5	Medium
4	Cotyledon shape	1	Narrow elliptic			7	Dense
		2	Elliptic	11	Fruit shape	1	Rounded
		3	Broad elliptic			2	Elongated
5	Density of foliage in plant	3	Sparse			3	Elliptic
		5	Medium	12	Intensity of brown color in fruit	3	Light
		7	Dense			5	Medium
6	Coloration in foliage	1	Yellowish green			7	Dark
		3	Green				
		5	Dark green				

**Table 3 biology-13-00866-t003:** Fifty percent seedling, flowering, and fruit setting days of the first top 10 coriander genotypes with cultivars.

Genotypes/Cultivars	50% SD	G/C	%50 FD	G/C	%50 FSD
Ames 18574	24.37 S	Ames 18570	52.40 mn	Ames 18567	61.40 gh
Ames 18568	24.97 S	Ames 18563	52.40 mn	Ames 18561	61.40 gh
Ames 23624	25.57 S	Ames 19089	53.00 lmn	Ames 18572	62.40 fg
Ames 18569	25.97 Q–S	Ames 18573	53.40 klm	Ames 18569	62.40 fg
Ames 18570	26.97 P–S	Ames 18572	53.40 klm	Ames 12778	62.40 fg
Ames 18595	29.37 O–Q	Ames 18568	53.40 k–n	Ames 18571	62.40 fgh
Ames 18580	29.37 O–Q	Ames 10234	53.4 klm	Ames 10234	63.40 efg
Ames 18575	29.37 O–Q	Ames 18588	54.00 j–m	PI 274290	65.00 d–g
Ames 18588	29.37 O–R	Ames 12778	54.40 ı–m	Ames 23626	65.40 c–g
Ames 23618	29.57 N–Q	Ames 23623	54.60 h–m	Ames 23618	65.40c–g
Arslan	36.57 B–F	Arslan	56.20 d–m	Arslan	67.17 Z–g
Erbaa	36.57 B–F	Erbaa	55.33 f–m	Erbaa	78.50 G–R
Gamze	36.57 B–F	Gamze	61.50 N–V	Gamze	75.67 K–Y
Gürbüz	33.17 F–N	Gürbüz	59.17 S–f	Gürbüz	73.17 O–c
Pelmus	33.33 F–M	Pelmus	59.00 T–g	Pelmus	78.50 G–R

Differences between means indicated with the same letter are not significant. G/C: genotypes/cultivars, SD: seedling days, FD: flowering days, FSD: fruit setting days.

**Table 4 biology-13-00866-t004:** The first top 10 genotypes for plant height, branch number, umbel number, and umbellet number values in different origin coriander genotypes and cultivars.

G/C	PH	G/C	BN	G/C	UN	G/C	UmbltN
PI 174129	84.30 A	PI 193769	15.10 A	Ames 13900	72.68 A	Ames 13900	380.19 A
Ames 21655	83.24 AB	Ames 13900	12.74 B	Ames 13899	62.61 B	PI 193769	315.31 B
Ames 29172	81.10 A–C	Ames 23623	11.62 BC	PI 193769	52.41 C	Ames 13899	200.19 C
Ames 18578	77.52 A–D	PI 269472	11.51 B–D	Ames 10235	43.61 CD	Ames 24909	156.79 CD
Ames 20046	76.24 A–E	Ames 23642	11.42 B–D	Ames 24909	40.57 DE	Ames 10235	153.19 C–E
Ames 20046	76.24 A–E	Ames 13899	11.14 B–E	Ames 20046	35.65 DEF	Ames 18595	148.55 D–F
PI 193769	75.90 A–F	Ames 18595	11.06 B–F	Ames 18568	34.41 D–G	Ames 19089	138.91 D–G
Ames 18581	73.92 A–G	Ames 21655	11.02 B–F	PI 664512	32.77 E–H	Ames 23642	130.19 D–H
Ames 23621	73.84 A–G	Ames 18581	10.66 B–G	Ames 23642	32.17 E–I	PI 170320	126.11 D–H
Ames 19089	73.64 A–G	PI 174129	10.50 B–H	Ames 20048	32.05 E–I	Ames 24907	98.79 G–P
Pelmus	58.52 G–c	Pelmus	6.33 T–l	Pelmus	11.47 Y–q	Pelmus	38.57 V–l
Gürbüz	59.37 G–b	Gürbüz	7.50 L–d	Gürbüz	12.90 T–p	Gürbüz	53.43 O–k
Gamze	65.63 C–R	Gamze	7.63 L–b	Gamze	18.87 N–d	Gamze	68.20 M–e
Erbaa	70.63 A–L	Erbaa	7.47 M–d	Erbaa	20.47 K–Z	Erbaa	85.10 H–Y
Arslan	49.45 R–m	Arslan	7.17 N–e	Arslan	11.97 Y–q	Arslan	40.27 T–l

Differences between means indicated with the same letter are not significant. G/C: genotypes/cultivars, PH: plant height, BN: branch number, UN: umbel number, UmbltN: umbellet number.

**Table 5 biology-13-00866-t005:** The first top 10 genotypes for 1000 fruit weight, fruit yield, biological yield, and harvest index of different coriander genotypes and cultivars.

G/C	1000 FW	G/C	FY	G/C	BY	G/C	HI
Ames 18569	21.49 A	Ames 13900	9.58 A	Ames 13900	50.78 A	Ames 23642	73.36 A
Ames 18572	20.74 A	Ames 10234	6.58 B	PI 193769	26.21 B	Ames 24921	48.52 B
Ames 18585	19.42 AB	Ames 18573	6.39 BC	Ames 18573	24.34 BC	Ames 18595	46.27 BC
Ames 18571	17.62 BC	Ames 18581	6.13 B–D	Ames 18581	24.30 BC	Ames 18572	45.29 B–D
Ames 23635	17.06 CD	Ames 24909	6.04 B–E	Ames 13899	23.86 B–D	PI 478378	43.32 B–E
Ames 23641	16.89 CD	Ames 10235	6.04 B–E	Ames 10235	22.12 B–E	Ames 19089	42.45 B–F
Ames 18596	16.50 CD	Ames 13899	5.92 B–F	PI 172808	20.78 B–F	Ames 18588	42.14 B–G
Ames 23614	16.38 C–E	Ames 27391	5.84 B–G	Ames 10234	20.42 B–G	Ames 18571	41.79 B–H
PI 274290	15.24 D–F	PI 172808	5.17 B–H	Ames 23640	19.46 B–H	Ames 18585	41.66 B–H
PI 193769	15.11 D–F	PI 633685	4.85 B–I	PI 170319	17.61 B–I	Ames 21108	41.50 B–I
Pelmus	7.29 ı–2	Pelmus	1.41 N–e	Pelmus	5.61 O–d	Pelmus	25.42 V–o
Gürbüz	10.33 N–a	Gürbüz	2.84 H–c	Gürbüz	8.22 I–d	Gürbüz	34.39 E–V
Gamze	9.08 T–l	Gamze	2.98 H–b	Gamze	9.86 I–c	Gamze	31.46 J–e
Erbaa	7.54 h–z	Erbaa	3.78 B–Q	Erbaa	13.87 E–Q	Erbaa	26.56 Q–o
Arslan	17.37 BC	Arslan	2.99 G–b	Arslan	8.90 I–d	Arslan	34.37 E–V

Differences between means indicated with the same letter are not significant. G/C: genotypes/cultivars, 1000 FW: 1000 fruit weight, FY: fruit yield, BY: biological yield, HI: harvest index.

**Table 6 biology-13-00866-t006:** Essential oil contents of different origin high-yielding coriander genotypes.

G/C	EOC (%*v*/*w*)	G/C	EOC (%*v*/*w*)	G/C	EOC (%*v*/*w*)
PI 669963	0.40 Y–d	Ames 27771	0.24 ıj	Ames 18588	0.41 V–c
PI 669959	0.50 P–S	Ames 27770	0.24 ıj	Ames 18587	0.61 J–M
PI 664512	0.50 P–S	Ames 27392	0.49 Q–T	Ames 18585	0.36 b–f
PI 633685	0.60 K–N	Ames 27391	0.94 F	Ames 18583	0.56 M–P
PI 531296	1.20 E	Ames 25170	0.39 a–e	Ames 18582	0.31 fgh
PI 531293	1.65 C	Ames 24926	0.49 Q–T	Ames 18581	0.96 F
PI 502320	1.75 B	Ames 24923	0.24 ıj	Ames 18580	0.51 O–R
PI 483232	0.20 ıjk	Ames 24921	0.44 T–a	Ames 18578	0.31 fgh
PI 478378	0.80 GH	Ames 24917	0.49 Q–T	Ames 18577	0.66 IJK
PI 269472	0.10 lm	Ames 24909	0.14 l	Ames 18575	0.46 R–Y
PI 269470	0.50 P–S	Ames 24907	0.99 F	Ames 18574	0.56 M–P
PI 268378	0.51 O–R	Ames 23639	0.25 hı	Ames 18573	0.56 M–P
PI 256061	0.51 O–R	Ames 23635	0.34 efg	Ames 18572	0.30 gh
PI 196843	0.21 ıj	Ames 23634	0.69 I	Ames 18571	0.30 gh
PI 193770	0.46 R–V	Ames 23632	0.34 efg	Ames 18570	0.60 LMN
PI 193493	0.10 lm	Ames 23624	0.05 m	Ames 18569	0.35 d–g
PI 174129	0.41 U–b	Ames 23622	0.53 OPQ	Ames 18568	0.45 S–Z
PI 171592	0.21 ıj	Ames 23621	0.06 m	Ames 18567	0.35 d–g
PI 170320	0.81 G	Ames 23616	0.55 NOP	Ames 18566	0.65 I–L
Pelmus	0.60 LMN	Ames 23614	0.20 ıjk	Ames 18565	0.60 LMN
Gürbüz	0.35 c–g	Ames 21655	0.46 R–U	Ames 18564	0.80 GH
Gamze	0.55 M–P	Ames 20048	0.55 NOP	Ames 18563	0.80 GH
Erbaa	0.55 M–P	Ames 20046	0.40 Z–d	Ames 18561	0.20 jk
Arslan	0.15 kl	Ames 19089	1.50 D	Ames 18560	0.65 I–L
Ames 4998	0.65 I–L	Ames 18596	0.51 O–R	Ames 18559	0.75 H
Ames 23640	0.25 hı	Ames 18595	1.51 D	Ames 13900	1.86 A
Ames 29174	0.76 GH	Ames 18594	0.51 O–R	Ames 13899	0.65 I–L
Ames 29173	0.66 IJ	Ames 18592	1.16 E	Ames 10235	0.53 OPQ
Ames 29172	0.51 O–R	Ames 18591	0.76 GH	Ames 10234	0.40 Z–d
Ames 27870	0.31 fg	Ames 18590	0.66 IJK		
Ames 27772	0.56 MNO	Ames 18589	0.76 GH		

Differences between means indicated with the same letter are not significant. G/C: genotypes/cultivars, EOC: essential oil content.

**Table 7 biology-13-00866-t007:** Major essential oil compositions of selected high-yielding coriander genotypes.

G/C/EOCom	Linalool (%*v*/*v*)	Camphor (%*v*/*v*)	γ-Terpinene (%*v*/*v*)	*p*-Cymene (%*v*/*v*)	β-Pinene (%*v*/*v*)	Total
RT (min)	23.63	25.95	21.17	20.4	18.48	
PI 664512	6.48 YZ	6.32 F	8.34 H	5.07 A–D	1.03 N	25.15
PI 633685	6.45 YZa	4.35 N	6.56 S	3.31 C–F	1.17 M	29.09
PI 483232	10.60 Q	5.28 H	8.32 H	4.50 A–E	1.59 FG	21.19
PI 478378	7.80 ST	4.09 OP	6.92 P	4.37 A–F	1.94 C	28.20
PI 269472	13.92 M	3.75 RS	9.66 C	2.68 D–G	0.55 R	22.37
PI 193493	6.68 VY	4.28 NO	7.05 O	4.97 A–D	1.94 C	34.54
PI 174129	11.81 P	0.97 a	4.60 b	1.58 FG	0.98 N	29.05
PI 171592	6.42 YZa	4.55 LM	6.04 V	3.61 B–F	1.84 D	26.26
Pelmus	7.29 TU	11.18 A	8.68 G	4.13 A–F	0.88 O	26.46
Gürbüz	7.12 UV	4.40 LMN	6.24 T	3.65 B–F	0.74 P	29.18
Gamze	29.54 E	4.36 MN	7.88 J	3.26 C–F	0.25 V	42.24
Erbaa	15.29 K	3.80 QR	5.06 a	3.77 B–F	0.34 U	40.02
Arslan	5.89 abc	3.80 QR	6.57 S	5.30 A–D	1.26 L	30.56
Ames 29174	45.70 A	4.89 K	0.05 d	0.02 G	2.07 B	71.13
Ames 24921	25.81 F	4.57 L	7.69L	3.97 B–F	0.37 STU	54.97
Ames 24909	12.95 N	5.95 G	8.93F	6.24 AB	1.25 L	41.54
Ames 24907	30.25 D	6.83 D	15.60A	4.08 A–F	0.54 R	54.49
Ames 23640	7.74 ST	9.37 B	9.34 E	5.21 A–D	0.60 Q	19.75
Ames 23635	35.00 B	3.57 SV	5.07 a	1.92 EFG	0.36 TU	54.47
Ames 23634	19.23 I	3.39 VYZ	12.91 B	6.90 A	0.71 P	32.61
Ames 20048	14.55 L	0.05 d	7.81 JK	4.43 A–F	0.62 Q	35.65
Ames 19089	9.20 R	6.59 E	6.2 TU	3.65 B–F	0.83 O	31.93
Ames 18595	7.29 TU	5.04 JK	6.60 S	4.65 A–E	1.28 KL	27.17
Ames 18591	6.08 Z–c	5.08 IJK	7.51 M	4.90 A–D	1.55 GH	28.31
Ames 18590	5.56 c	3.72 RST	6.88 P	5.29 A–D	1.86 D	18.94
Ames 18587	21.40 G	3.63 R–U	5.57 Y	3.07 C–F	0.10 Z	38.76
Ames 18585	33.87 C	7.15 C	9.48 D	3.77 B–F	0.42 S	51.70
Ames 18581	7.87 S	3.71 RST	6.1 UV	4.48 A–E	1.15 M	24.75
Ames 18577	3.13 d	0.74 b	6.74 QR	5.51 A–D	1.79 E	19.48
Ames 18575	9.06 R	3.37 YZ	6.85 PQ	4.58 A–E	1.41 J	29.81
Ames 18573	5.73 bc	5.14 HIJ	5.15 a	3.31 C–F	1.47 I	21.97
Ames 18570	12.77 NO	5.24 HI	7.26 N	5.81 ABC	0.60 Q	36.26
Ames 18569	6.29 Y–b	3.97 PQ	5.65 Y	3.80 B–F	0.83 O	26.51
Ames 18568	20.68 H	6.06 G	9.52 D	5.54 ABC	0.39 ST	38.23
Ames 18566	9.09 R	3.49 U–Z	7.44 M	4.92 A–D	1.50 HI	24.97
Ames 18559	6.85 UVY	3.34 Z	5.37 Z	3.62 B–F	0.84 O	29.34
Ames 13900	7.06 UV	5.23 HIJ	7.76 KL	5.59 ABC	1.32 K	25.34
Ames 13899	9.62 R	4.14 OP	8.07 I	4.91 A–D	1.61 F	31.23
Ames 10235	12.35 OP	3.55 T–Y	4.28 c	2.98 C–F	2.61 A	26.63
Ames 10234	16.69 J	0.37 c	6.67 RS	3.83 B–F	0.19 Y	31.14

Differences between means indicated with the same letter are not significant. G/C: genotypes/cultivars, EOCom: essential oil composition, RT: retention time.

**Table 8 biology-13-00866-t008:** Fixed oil contents of the high-yielding different origin coriander genotypes.

G/C	FOC (%*v*/*w*)	G/C	FOC (%*v*/*w*)	G/C	FOC (%*v*/*w*)
PI 664512	25.70 S	Ames 24921	29.49 N	Ames 18577	16.31 h
PI 633685	24.98 U	Ames 24909	30.91 J	Ames 18575	13.72 k
PI 483232	25.35 T	Ames 24907	30.60 K	Ames 18573	28.86 O
PI 478378	22.11 b	Ames 23640	31.85 G	Ames 18570	31.39 I
PI 269472	25.89 R	Ames 23635	24.06 Z	Ames 18569	16.27 ı
PI 193493	20.35 c	Ames 23634	33.63 C	Ames 18568	10.35 n
PI 174129	20.21 d	Ames 20048	24.50 Y	Ames 18566	30.18 L
PI 171592	19.35 f	Ames 19089	28.56 P	Ames 18559	33.19 E
Pelmus	22.43 a	Ames 18595	30.04 M	Ames 13900	32.21 F
Gürbüz	33.93 B	Ames 18591	27.90 Q	Ames 13899	14.39 j
Gamze	33.27 D	Ames 18590	24.73 V	Ames 10235	10.22 o
Erbaa	20.20 d	Ames 18587	12.52 m	Ames 10234	34.03 A
Arslan	31.4 H	Ames 18585	13.56 l		
Ames 29174	19.50 e	Ames 18581	17.60 g		

Differences between means indicated with the same letter are not significant. G/C: genotypes/cultivars, FOC: fixed oil content.

**Table 9 biology-13-00866-t009:** Major fixed oil acids of the high-yielding coriander genotypes.

G/C-FOA	Petroselinic (C18:1n12)	Palmitic (C16:0)	Elaidic (C18:1n9t)	Behenic (C22:0)	Arachidic (C20:0)	Total
RT	20.94	17.73	21.82	28.02	24.42	
PI 664512	63.12 O	10.83 d	7.11 Y	7.11 N	3.48 K	91.65
PI 633685	61.17 Q	12.19 O	10.60 N	9.08 F	2.54 R	95.58
PI 483232	66.00 K	10.37 g	8.20 S	7.51 K	2.15 V	94.23
PI 478378	41.26 ı	10.97 c	27.27 B	7.49 K	2.20 U	89.19
PI 269472	46.56 f	14.97 D	18.23 G	5.13 V	2.99 M	87.88
PI 193493	58.08 S	12.80 K	6.94 Z	7.06 O	5.65 E	90.53
PI 174129	39.92 j	13.26 I	26.00 C	5.92 S	2.20 U	87.30
PI 171592	55.90 U	13.52 H	12.73 J	6.86 P	2.98 N	91.99
Pelmus	58.10 S	12.03 R	9.72 P	7.68 I	5.07 H	92.60
Gürbüz	71.65 G	14.86 E	4.65 e	*nd*	*nd*	91.16
Gamze	24.47 n	23.04 A	47.44 A	*nd*	*nd*	94.95
Erbaa	46.16 g	12.33 M	15.84 I	5.04 Z	9.86 C	89.23
Arslan	65.52 L	11.08 Z	11.05 L	6.03 R	1.23 f	94.91
Ames 29174	62.69 P	12.48 L	7.93 T	3.89 f	3.92 J	90.91
Ames 24921	49.66 d	12.15 P	23.49 D	7.02 O	2.51 S	94.83
Ames 24909	53.95 Z	14.48 F	17.32 H	5.61 T	1.70 c	93.06
Ames 24907	74.97 D	10.34 h	2.14 ı	9.24 E	*nd*	96.69
Ames 23640	67.73 I	11.54 U	4.43 f	12.56 A	1.08 g	97.34
Ames 23635	59.87 R	11.00 b	10.38 O	4.13 e	1.89 Z	87.27
Ames 23634	51.12 b	12.30 N	10.73 M	10.19 D	10.84 B	95.18
Ames 20048	51.36 a	10.21 ı	7.45 V	5.26 U	1.92 Y	76.20
Ames 19089	48.82 e	11.39 V	11.42 K	7.29 M	5.63 F	84.55
Ames 18595	46.58 f	10.38 g	22.32 E	4.19 d	5.16 G	88.63
Ames 18591	66.47 J	12.11 Q	*nd*	7.86 H	4.68 I	91.12
Ames 18590	39.38 k	7.22 l	6.13 c	7.66 I	2.23 T	62.62
Ames 18587	31.17 m	11.73 T	20.15 F	10.70 B	1.84 a	75.59
Ames 18585	64.31 N	10.79 e	4.08 g	4.15 e	1.59 d	84.92
Ames 18581	45.84 h	17.16 B	4.81 d	3.17 g	1.90 Z	72.88
Ames 18577	54.52 Y	*nd*	8.85 R	4.51 c	2.67 P	70.55
Ames 18575	50.22 c	13.28 I	6.68 a	4.54 c	1.83 b	76.55
Ames 18573	56.82 T	13.18 J	2.92 h	5.09 Y	7.94 D	85.95
Ames 18570	65.45 M	10.49 f	7.44 V	4.84 a	2.71 O	90.93
Ames 18569	73.23 F	7.13 m	1.55 j	7.58 J	3.30 L	92.79
Ames 18568	74.88 E	16.78 C	7.82 U	*nd*	*nd*	99.48
Ames 18566	36.53 l	11.06 a	7.12 Y	*nd*	13.33 A	68.04
Ames 18559	87.70 A	10.04 j	*nd*	*nd*	*nd*	97.74
Ames 13900	80.53 B	11.91 S	*nd*	*nd*	*nd*	92.44
Ames 13899	79.71 C	9.76 k	*nd*	7.42 L	*nd*	96.89
Ames 10235	68.80 H	11.33 Y	6.30 b	10.31 C	1.48 e	98.22
Ames 10234	55.06 V	13.77 G	9.32 Q	6.22 Q	2.62 Q	86.99

Differences between means indicated with the same letter are not significant. G/C: genotype/cultivar, FOA: fixed oil acid, RT: retention time, *nd*: not detected.

**Table 10 biology-13-00866-t010:** Correlation analysis results of the morphological and yield values of the different coriander genotypes.

Characters	50% FD	50% FSD	PH	BN	UN	UmblN	1000 FW	FY	BY	HI
50% SD	0.014	−0.135	−0.173	−0.089	−0.011	−0.02	0.076	−0.009	0.022	−0.068
50% FD		0.246 **	0.174	0.212 *	0.164	0.182	−0.117	0.109	0.1	−0.063
50% FSD			0.66 **	0.413 **	0.252 **	0.347 **	−0.475 **	0.321 **	0.246 **	0.039
PH				0.67 **	0.46 **	0.529 **	−0.348 **	0.459 **	0.46 **	0.038
BN					0.752 **	0.752 **	−0.207 *	0.53 **	0.574 **	0.027
UN						0.902 **	−0.243 **	0.652 **	0.724 **	−0.024
UmblN							−0.267 **	0.648 **	0.782 **	−0.042
1000 FW								−0.053	−0.06	0.204 *
FY									0.856 **	0.073
BY										−0.161

*: Significant at 5%, **: Significant at 1%, SD: seedling days, FD: flowering days, FSD: fruit setting days, PH: plant height, BN: branch number, UN: umbel number, UmbltN: umbellet number, 1000 FW: 1000 fruit weight, FY: fruit yield, BY: biological yield, HI: harvest index.

## Data Availability

Data are contained within the article.

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
