# Peer review of "Genetic Diversity, Analysis of Some Agro-Morphological and Quality Traits, and Utilization of Plant Resources of Coriander (Coriandrum sativum) Supported with Cluster and Multivariate Analyses"

_biology, 2024, doi:10.3390/biology13110866_

Round 1
Reviewer 1 Report
Comments and Suggestions for Authors
The authors have done a large amount of work to study the morphology, size, yield, and biochemical composition of the fruits of 119 coriander accessions of different geographical origins. This material is useful for breeding and seed production.
Comments on the text.
line 133 «In the experiment, the fruits were sown on April 30, 2019, with an equal amount of fruit in each row» does not indicate how many fruits of each accession were used and what the germination rate was. In methods, the authors did not write how they calculated the Harvest Index.
Line 275 and further. A simple umbel that is part of a compound umbel in Umbelliferae is called an “umbellule” or “umbellet” but not an “umbellаte”. (e.g., G.M. Plunkett et al., 2018. Apiaceae. In: Ed. K. Kubitzki. The Families and Genera of Vascular Plants. V. 15. Springer International Publishing AG, part of Springer Nature, 2018. P. 9 – 206).
The authors mixed up the features “umbellules per umbel” and “umbellules per plant”.
The Table S3 (and probably other tables) contains a lot of typos and errors, that were carried over into the main text of the article and the Tables 4 and 5. They make doubt on the reliability of the data in the table S3 and the correct presentation of the results. The authors should carefully check the tables against the actual measurements and correct the text.
The color scale is not deciphered in Figures 1, 2, and 3. Line 201 - Peterson (1994) is missing from the list of references.
Author Response
Dear Dr. Reviewer-1,
We thank your priceless suggestions which we believe will improve the manuscript. Many thanks for your constructive manner and detailed review of our manuscript. The corrections were indicated as colored text parts in red for your comments. We believe the manuscript is in better shape now. We hope the corrections and explanations would be sufficient.
Comment 1: line 133 «In the experiment, the fruits were sown on April 30, 2019, with an equal amount of fruit in each row» does not indicate how many fruits of each accession were used and what the germination rate was. In methods, the authors did not write how they calculated the Harvest Index.
Response 1: Page: 4 and 6 Line number: 128-131, and 201-203
Please, we thank the reviewer for his/her time and evaluating of our manuscript. The sentence was corrected as “In the experiment, the fruits were sown on April 30, 2019, and sowing was conducted with an excessive number of fruits (approximately 50), enabling thinning to be per-formed 10-15 days after emergence to achieve the desired plants in each row.” Also, in methods, the calculation of harvest index was given as “The harvest index was calculated by dividing the biological yield by the fruit yield and multiplying the result by 100 as using following formula.
Harvest index= (Fruit yield/Biological yield) ×100.”
Comment 2: Line 275 and further. A simple umbel that is part of a compound umbel in Umbelliferae is called an “umbellule” or “umbellet” but not an “umbellаte”. (e.g., G.M. Plunkett et al., 2018. Apiaceae. In: Ed. K. Kubitzki. The Families and Genera of Vascular Plants. V. 15. Springer International Publishing AG, part of Springer Nature, 2018. P. 9 – 206).
Response 2: Page: 6, 9, 10 Line number: 193, 194, 307-353
Please, you can see that the umbellate number was changed as umbellet during the manuscript. We are so sorry to write umbellate per umbel. We wrote it from some manuscript as “1- Effect of various doses of nutrients on growth and yield parameters of coriander; 2- Qualitative and yield characters in coriander genotypes; 3- Study on genetic variability in germplasm of coriander (Coriandrum sativum L.); 4- Performance of different coriander varieties for seed yield; 5- Correlation study and path analysis of coriander (Coriandrum sativum L.) for yield and its attributes in mid hills of Uttarakhand”. Also, we corrected it as “umbellet per plant”.
Comment 3: The authors mixed up the features “umbellules per umbel” and “umbellules per plant”.
Response 3: Page: 6, 9, 10 Line number: 193, 194, 307-353
Thank you very much for your correction. Please, you can see that the features “umbellules per umbel” and umbellules per plants” were changed as “umbel number per plant” and “umbellet number per plant” during the manuscript.
Comment 4: The Table S3 (and probably other tables) contains a lot of typos and errors, that were carried over into the main text of the article and the Tables 4 and 5. They make doubt on the reliability of the data in the table S3 and the correct presentation of the results. The authors should carefully check the tables against the actual measurements and correct the text.
Response 4:
Dear reviewer, thank you very much for your correction and suggestions. Please you can see that all Tables with in the text or supplementary tables were controlled and the needed corrections were done.
Comment 5: The color scale is not deciphered in Figures 1, 2, and 3.
Response 5: Page: 21, 22, 23 Line number: 695, 696, 699-702, 715, 716
Please, you can see that the color scale was deciphered for Figures 1, 2 and 3 as “Green to white to red colors show the difference among the examined properties and genotypes. Green color indicates lower data values, white color shows average value and red represent higher data values.”, “Blue to grey to red colors show the difference among the examined properties and genotypes. Red and light shades of red colors indicate higher data values, and light shades of blue to blue colors represent lower data values.” and “From orange to red (1-2) colors indicate higher data values, and from light shades of blue (0-1) to blue (-1-0) colors represent lower data values”, respectively.
Comment 6: Line 201 - Peterson (1994) is missing from the list of references.
Response 5: Page: 27, Line number: 849-850
Please, you can see that the The reference Peterson (1994) was added to References part as “Peterson, R.G. Agricultural Field Experiments: Design and Analysis (1st ed.). CRC Press. Marcel Dekker Inc., Boca Ra-ton, Florida, USA, 1994, 428 Pages.”
Best regards,
Reviewer 2 Report
Comments and Suggestions for Authors
In the manuscript, the coriender genotypes from different regions have been evaluted based on their different characteristics for breeding and climate change aspects. Manuscripts inludes valuable results but it needs to improve in some aspects as indicated below.
1) Line 48: it would be better to use Has been over instead of crossed.
2) the terminology must be same throught the manuscript. eg. plant material/genotype (Line 115/116). It should be used genotypes as indicated the other parts.
3) Seed must be used instead of fruit. thorught the manuscript. eg: line 133, 223, 320, 322 etc..
4) Line 188 ''Morphological characteristics'' should be used instead of ''UPOV criteria examined''
5) In the material and method part, there is no explanation for how those parameters (50% seedling day, 50% flower day, 50% fruit setting day, harvest index, biological yield) measured or evaluated.
6) Based on table 3,there is no enough discussion on the why the earliest genotypes in days to 50% seedling had not same the earliest in days to 50% flowering and fruit set. Beause it is expected that early germinate genotypes tend to early flowering and fruit set. There is no explanation on this issue.
7) It would be better to indicate the correlation analysis of the examined chracteristics.
Comments on the Quality of English Language
English of the manuscript is clear and understandable.
Author Response
Dear Dr. Reviewer-2,
We thank your priceless suggestions which we believe will improve the manuscript. Many thanks for your constructive manner and detailed review of our manuscript. The corrections were indicated as colored text parts in red for your comments. We believe the manuscript is in better shape now. We hope the corrections and explanations would be sufficient.
Comment 1: Line 48: it would be better to use Has been over instead of crossed.
Response 1: Section: Introduction, Page: 2, Line number: 44
We thank the reviewer 2 for his/her time and evaluating of our manuscript. Also thanks to his/her suggestions and corrections. Please, you can see that the crossed word was deleted and corrected as “has been over”.
Comment 2: the terminology must be same through the manuscript. eg. plant material/genotype (Line 115/116). It should be used genotypes as indicated the other parts.
Response 2: Section: Materials and Methods, Page: 3, Line number: 111, 114
Please you can see that the plant material was corrected as “genotypes” and “genotypes and cultivars”.
Comment 3: Seed must be used instead of fruit. thought the manuscript. eg: line 133, 223, 320, 322 etc...
Response 3:
Dear reviewer, thank you very much for your correction. But we are so sorry to give negative response to your suggestion. Because the coriander has fruit not seed. We also controlled from different published manuscripts and saw the coriander fruit. Because previous studies reported that coriander fruits as “Actually fruits not seeds, these are obtained from the plant which starts flowering in June yielding round fruits consisting of two pericarps. The fruits are 2–8 mm in length from which the oil is obtained once they are dried (Zeb, 2016).” So, we wrote fruit instead of seed.
Comment 4: Line 188 ''Morphological characteristics'' should be used instead of ''UPOV criteria examined''
Response 4: Material and Methods Section, Page: 5, 6, line number: 183-203
Please, you can see that the morphological characteristics were added together with the UPOV criteria examined. Because morphological and UPOV criteria are different in this study and the needed reference was given for UPOV criteria. The paragraphs were added as “The morphological characteristics in the study were determined as follows: Days to 50% seedlings, flowering and fruit setting were recorded as the number of days from sowing until 50% of the plants in a net plot produced seedlings, flowers and fruits as determined by visual observation, respectively. Plant height was measured in centi-meters (cm) for ten randomly selected plants, from ground level to the apex, at the time of physiological maturity in the net plot area. The average number of primary branches that emerged directly from the main shoot was counted for ten randomly selected plants at physiological maturity. The number of umbels from 10 randomly selected plants in each row was counted, and the average number of umbels per plant was calculated. The number of umbelletes from 10 randomly selected plants in each row was counted, and the average number of umbelletes per plant was calcu-lated. Thousand fruit weight was calculated as from the portion of fruits separated as pure seed from each row, four samples of 100 fruits were weighed using a precision scale. The average weight from these four measurements was then multiplied by 10 to determine the thousand grain weight in grams. After harvest, the fruits from the threshed plants were weighed, and the fruit yield per plant was recorded in g/plant. Biological yield per plant (g) was calculated on the dry weight of the harvested plants. The harvest index was calculated by dividing the biological yield by the fruit yield and multiplying the result by 100 as using following formula.
Harvest index= (Fruit yield/Biological yield) ×100.”
Comment 5: In the material and method part, there is no explanation for how those parameters (50% seedling day, 50% flower day, 50% fruit setting day, harvest index, biological yield) measured or evaluated.
Response 5:
Please you can see that the needed response was given in Comment 4.
Comment 6: Based on table 3, there is no enough discussion on the why the earliest genotypes in days to 50% seedling had not same the earliest in days to 50% flowering and fruit set. Because it is expected that early germinate genotypes tend to early flowering and fruit set. There is no explanation on this issue.
Response 6: Results and discussion part, Page: 8, line number: 267-268
Dear reviewer, thank you very much for your correction and suggestion. You are right. Please you can see that the needed information was added to our manuscript as “In fact, a positive association between days to seedling (50%) and days to flowering (50%) is expected. However, in our study, the early seedling days could not show a meaningful and significant relationship the early flowering days. Differences in early seedling days and early flowering days among coriander genotypes have been attributed to variations in temperature, light intensity, and competition for resources between reproductive and vegetative tissues [25]. In addition, genes for phenology and plant development, their interactions with each other and the environment may affect the late or early flowering [26]. It was determined that 50% flowering and 50% fruit setting days were found partially similar among the genotypes. The differences can be explained with reducing pollination and fruit-setting for an insect pollinated coriander genotypes that flowered earlier [27].” Also, the statistical analysis method as explained in the statistical part for the Augmented Trial Design is different from the other trial design. The values for the examined properties can increase or decrease in each block depending on the ANOVA analysis results for used repeated cultivars. So, the data values for seedling days, flowering days and fruit setting days can show differences.
Comment 7: It would be better to indicate the correlation analysis of the examined characteristics.
Response 7: Results Section, Page: 24-25, Line number: 733-763
Dear reviewer, thank you very much again for your suggestion. We added the correlation analysis which was conducted morphological and yield properties and the results were discussed. Please you can see that the results on correlation analysis were given as “The correlation analysis was conducted to determine the relationship among the morphological and yield values. A total of 29 correlations were identified among the examined characteristics, with 22 showing highly significant positive correlations (Table 10). The days to 50% fruit setting had the most correlations, interacting with seven of the characteristics studied. The highest significant positive correlation was found between the number of umbels and the number of umbelletes, with a correlation coefficient of r=0.902**. This was followed by the correlation between fruit yield and biological yield, which had a coefficient of r=0.856**. It was clearly reported that the highly significant positive correlation was found between the flowering and fruit setting days. Positive correlations were also observed between the days to 50% flowering and branch number, as well as between thousand fruit weight and harvest index. Highly significant negative correlations were noted between thousand fruit weight and days to 50% fruit setting, plant height, umbel number, and umbelletes. Additionally, two positive correlations were found: one between days to 50% flowering and branch number, and another between thousand fruit yield and harvest index. A negative correlation was detected between branch number and thousand fruit yield, with a coefficient of r=−0.207*.” with Table 10.
General comment: We revised English language of the our manuscript carefully.
Best regards,